# Core Optimization for Extending the Graphite Irradiation Lifespan in a Small Modular Thorium-Based Molten Salt Reactor

Xuzhong Kang [1], Guifeng Zhu [1,2,*], Jianhui Wu [1,2], Rui Yan [1,2], Yang Zou [1,2] and Yafen Liu [1]

1   Shanghai Institute of Applied Physics, Chinese Academy of Sciences, Shanghai 201800, China
2   University of Chinese Academy of Sciences, Beijing 100049, China
*   Correspondence: zhuguifeng@sinap.ac.cn; Tel.: +86-02139191017

**Abstract:** The lifespan of core graphite under neutron irradiation in a commercial molten salt reactor (MSR) has an important influence on its economy. Flattening the fast neutron flux ($\geq 0.05$ MeV) distribution in the core is the main method to extend the graphite irradiation lifespan. In this paper, the effects of the key parameters of MSRs on fast neutron flux distribution, including volume fraction (VF) of fuel salt, pitch of hexagonal fuel assembly, core zoning, and layout of control rod assemblies, were studied. The fast neutron flux distribution in a regular hexagon fuel assembly was first analyzed by varying VF and pitch. It was demonstrated that changing VF is more effective in reducing the fast neutron flux in both global and local graphite blocks. Flattening the fast neutron flux distribution of a commercial MSR core was then carried out by zoning the core into two regions under different VFs. Considering both the fast neutron flux distribution and burnup depth, an optimized core was obtained. The fast neutron flux distribution of the optimized core was further flattened by the rational arrangement of control rod channels. The calculation results show that the final optimized core could reduce the maximum fast neutron flux of the graphite blocks by about 30% and result in a more negative temperature reactivity coefficient, while slightly decreasing the burnup and maintaining a fully acceptable core temperature distribution.

**Keywords:** molten salt reactors; fast neutron flux distribution; graphite lifespan; burnup

## 1. Introduction

The molten salt reactor (MSR) is the only concept among the Generation IV reactors that adopts liquid fuel and offers many unique advantages in terms of safety, nuclear fuel utilization, economy, and nonproliferation [1,2]. MSRs have been studied at Oak Ridge National Laboratory (ORNL) since the 1950s [3]. Over the past seven decades, a variety of MSR types have been proposed, including chloride fast reactors [4,5], fluoride thermal reactors [6–12], and fluoride fast reactors [13,14]. Among these, the MSRs utilizing fluoride salt as the primary coolant and graphite blocks as the moderator are more technologically mature [15] and relatively easier to commercialize.

As the structural material of the MSR core, graphite blocks serve multiple roles: they act as the neutron moderator, provide channels for fluid flow, and maintain the mechanical stability of the reactor core. However, the structure of the graphite block undergoes deformation when irradiated by neutron flux. The graphite block initially shrinks and then gradually expands. The lifespan of the graphite block under neutron flux irradiation is generally defined as the period during which the graphite block shrinks and then expands back to its original volume. The irradiation lifespan of the graphite blocks in the core is mainly determined by fast neutrons with energy greater than 0.05 MeV [8]. Based on the irradiation behavior of small-sized graphite specimens, a permissible fast neutrons (>0.05 MeV) exposure for a MSR graphite block is about $3 \times 10^{22}$ (n/cm$^2$) [16]. Since 0.05 MeV is not a typical cutoff energy for fast neutrons, it is hereby clarified that the cutoff energy for fast neutrons is considered to be 0.05 MeV in this paper. Table 1 shows

the thermal power, maximum fast neutron flux of graphite blocks, designed irradiation lifespan of the core graphite, and volume fraction (VF) of fuel salt in the active core for some commercial conceptual MSRs.

**Table 1.** Irradiation lifespan of the core graphite for some commercial conceptual MSRs (the values in the table are derived from relevant references).

| MSR | Thermal Power (MW) | Maximum Fast Neutron Flux (>0.05 MeV) of Graphite Blocks ($cm^{-2}s^{-1}$) | Irradiation Lifespan of the Core Graphite (Years) | VF of Each Region in the Active Core from Inner to Outer |
| --- | --- | --- | --- | --- |
| MSBR [8] | 2250 | $3.5 \times 10^{14}$ | 4 | 13.2% |
| DMSR [17] | 1000 | $3.9 \times 10^{13}$ | 30 (the load factor is 0.75) | 34%, 32%, 26%, |
| MSR-2R [9] | 450 | $4.2 \times 10^{13}$ | 30 (the load factor is 0.75) | 27.3%, 23.4% |
| FUJI-U3 [18] | 450 | $4.1 \times 10^{13}$ | 30 (the load factor is 0.75) | 34%, 32%, 26%, |
| IMSR [19] | 400 | — | 5–7 | 13.7%, 15%, 16.3% |

As shown in Table 1, the irradiation lifespan of core graphite depends on the maximum fast neutron flux of graphite blocks. Currently, most graphite-moderated MSRs adopt core structures with the graphite blocks arranged regularly. However, the fast neutron flux within the core is inhomogeneously distributed, causing the graphite blocks located in the central region of the core to reach their irradiation lifespan prematurely, resulting in most of the graphite blocks in the core not being fully utilized. Additionally, the non-uniform distribution of fast neutron flux results in different deformation rates of the graphite blocks located at various positions within the core. This significantly affects the flow channels, subsequently influencing the heat transfer between the graphite and the fuel salt. Therefore, flattening the fast neutron flux distribution could not only prolong the irradiation lifespan of the graphite blocks but also improve the performance of thermal hydraulics.

Previously, the studies [17–20] have investigated flattening the fast neutron flux distribution in MSR cores by zoning the core with different volume fractions (VFs) of fuel salt. In DMSR [17], the core was divided into two regions, with the VFs for the inner and outer regions set at 20% and 12.9%, respectively. The diameter and height of the DMSR core were both 8.3 m. Referring to the core design of DMSR, the two- (MSR-2R) and three-region (FUJI-U3) design of FUJI cores [21] were carried out by Honma et al. [9] and Mitachi et al. [18], respectively. The diameter and height of both MSR-2R and FUJI-U3 cores were 6 m and 2.2 m, respectively. The VFs of the inner and outer regions of the MSR-2R core were 27.3% and 23.4%, respectively, and the VFs of the inner, intermediate, and outer regions of the FUJI-U3 core were 34%, 32%, and 26%, respectively. The irradiation lifespan of graphite blocks in the MSR-2R and FUJI-U3 cores was 30 years. However, these cores were designed with a large volume and low average power density of the fuel salt, which would significantly affect the economics of DMSR, MSR-2R, and FUJI-U3 as commercial reactors. Based on the designs of the DMSR, MSR-2R, and FUJI-U3 cores, Terrestrial Energy of Canada proposed the IMSR [19] to improve the economy of commercial MSRs. The diameter and height of the IMSR core were 3.5 m and 4 m, respectively, and the core was divided into three regions. The VFs of the inner, intermediate, and outer regions of the IMSR core were about 13.7%, 15%, and 16.3%, respectively.

Based on the above discussion, it can be concluded that those studies on flattening the fast neutron flux distribution in MSR cores have focused on core zoning. Furthermore, significant variations exist in the VF among various designed MSR cores, resulting in distinct energy spectra and fast neutron flux distributions within the core. The determination of VF appears to lack clear regulatory guidance. The current studies not only failed to consider the flux distribution within individual assemblies in the core, but also neglected the influence on other important core parameters, such as burnup depth, power distribution, and temperature reactivity coefficients, which may have adverse effects on these

parameters. Additionally, these studies did not consider the impact of the layout of control rod assemblies on the fast neutron flux distribution.

In this paper, based on the design of the small modular thorium-based molten salt reactor (sm-TMSR) [11] core, the effects of VF and pitch of hexagonal fuel assemblies, core zoning, and layout of control rod assemblies on fast neutron flux distribution were studied. The sm-TMSR core is optimized from the perspective of flattening the fast neutron flux distribution and burnup. Section 2 introduces the calculation models and methods. Sections 3 and 4 demonstrate the calculation results of the fuel assembly and core, respectively. Section 5 presents the conclusions of this paper.

## 2. Models and Methods of Calculation

### 2.1. Fuel Assembly and Corresponding Core Model

The sm-TMSR (150 MWt) is designed as a thorium converter and in situ burner driven by low-enriched uranium. It can be utilized for power generation, high-temperature hydrogen production, and the manufacture of radioactive medical isotopes, among other applications. Another important goal of the sm-TMSR is to achieve the utilization of thorium, which is expected to provide more than 40% of the energy output. This reactor serves as a commercial demonstration unit, and its design philosophy [11] includes the following aspects: (1) utilization of mature technologies and experiences; (2) designing the primary loop as a compact reactor module that can be replaced to address long-term material irradiation concerns; (3) adoption of online refueling and offline batch reprocessing mode; and (4) implementation of passive safety design principles.

The fuel assembly is the basic element that comprises the core. According to the calculations and analyses of references [12,22], an optimum fuel utilization and temperature reactivity coefficient can be achieved by setting VF to be about 10% in a single fuel assembly. As shown in Figure 1, the fuel assembly of the sm-TMSR core is a central cylindrical channel assembly with a regular hexagonal cross-section. The pitch of the regular hexagon is 18 cm, whereas the height of the assembly is 320 cm. The diameter of the fuel salt channel is 6 cm, and the VF is equal to 10.08%.

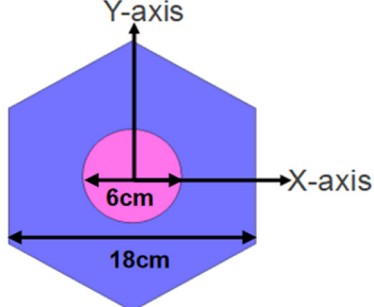

**Figure 1.** Fuel assembly model.

The sm-TMSR is currently in the pre-conceptual design stage. The core, depicted in Figure 2, serves as the benchmark core, and its main design parameters are detailed in Table 2. The active core has a diameter of 3 m and a height of 3.2 m, with the average power density of the fuel salt reaching approximately 66 MW/m$^3$. The fuel salt composition consists of LiF-BeF$_2$-UF$_4$-ThF$_4$, with the abundance of $^7$Li and $^{235}$U at 99.996 wt% and 19.75 wt%, respectively.

As can be seen from Table 2, the core will require replacement six times throughout the lifespan of the sm-TMSR. The flattening of fast neutron flux distribution in the core was studied to reduce the frequency of the sm-TMSR core replacement.

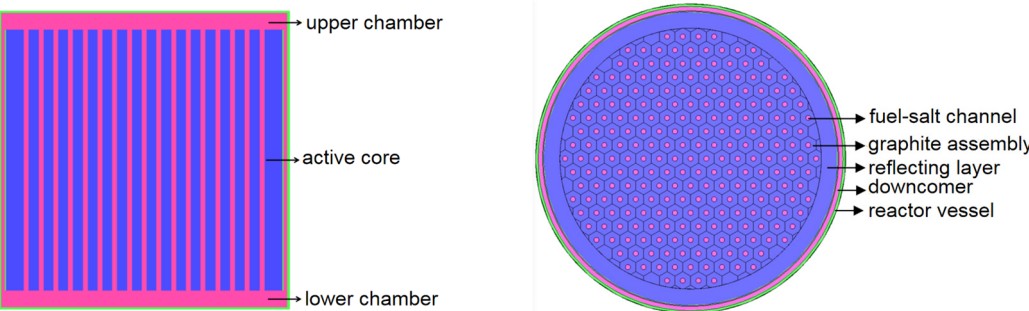

**Figure 2.** Core model of sm-TMSR.

**Table 2.** Main parameters of sm-TMSR core.

| Parameters | Value |
|---|---|
| Thermal power | 150 MW |
| Active core size | $\Phi$ 300 cm $\times$ 320 cm |
| Volume fraction of fuel salt | 10.08% |
| Thickness of reflector | 20 cm |
| Thickness of downcomer | 5 cm |
| Thickness of vessel | 3 cm |
| Thickness of upper chamber | 20 cm |
| Thickness of lower chamber | 20 cm |
| Core inlet/outlet temperature | 650 °C/700 °C |
| $^{235}$U enrichment | 19.75 wt% |
| Initial load of uranium | 1000 kg |
| Initial load of thorium | 5600 kg |
| Design life of the reactor | 60 years |
| Design life of the core | 10 years |

*2.2. Two-Region Core Design*

In this paper, the fast neutron flux distribution is calculated and analyzed by using the KENO module in SCALE6.1 [23]. The burnup calculation is carried out by using the MOBAT [24], which was developed to simulate the processes of online refueling, online removal of fission gas, and noble metal fission products in MSRs.

For the assembly model, the boundary condition of specular (mirror) reflection is adopted, and the fuel salt composition remains fixed in the calculation. Two scenarios were considered to modify the model depicted in Figure 1: (1) changing the fuel channel diameter of the regular hexagonal fuel assembly while fixing its pitch and (2) changing the pitch while fixing the VF of the fuel assembly.

Regarding the calculation of the core, a two-region core design is adopted to flatten the fast neutron flux distribution. The outer region remains unchanged while varying the VF or pitch of the fuel assemblies in the inner region. The layouts for the inner region (the blank area) are illustrated in Figure 3.

Figure 3 considers six configurations. The inner region of configuration I includes the center fuel assembly (marked as 0) and the first circle of fuel assemblies (marked as 1); the inner region of configuration II includes 0, 1, and 2 circles, and so on; the inner region of configuration VI includes 0, 1, 2, 3, 4, 5, and 6 circles.

In the next section, a single fuel assembly calculation and analysis was carried out to determine whether it would be better to change the VF or the pitch. The values of VF and pitch range from approximately 1% to 63% and from 6 cm to 40 cm, respectively.

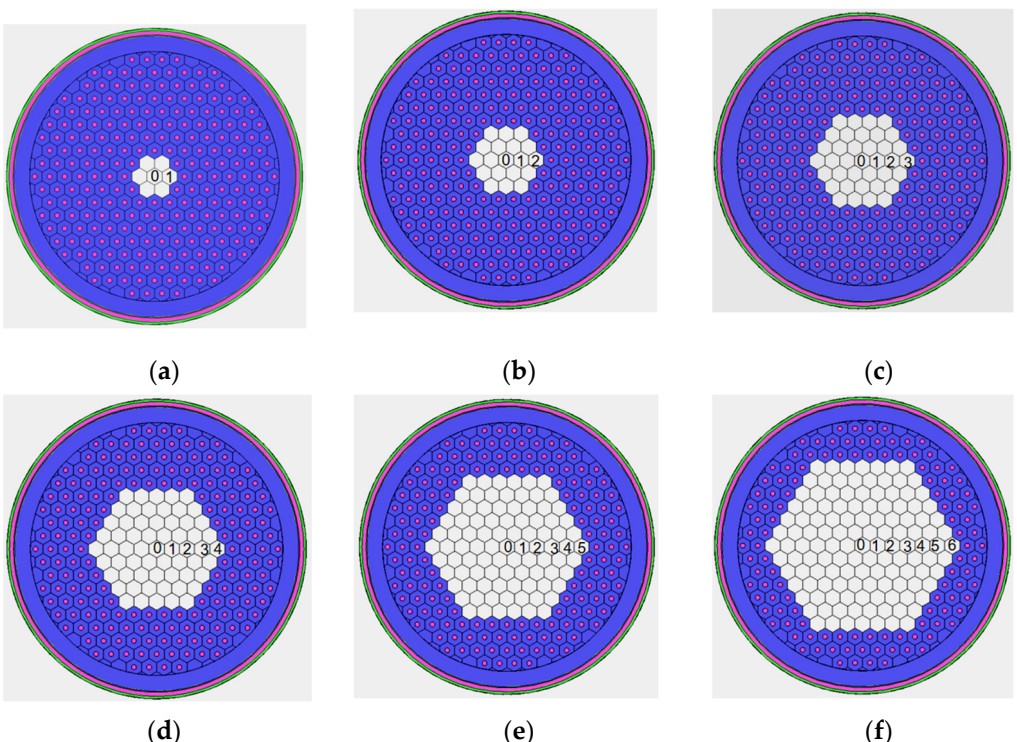

**Figure 3.** All configurations of two-region core: (**a**) Configuration I; (**b**) Configuration II; (**c**) Configuration III; (**d**) Configuration IV; (**e**) Configuration V; (**f**) Configuration VI.

## 3. Calculation Results of the Fuel Assembly

In the Monte Carlo calculation, the number of particles, non-active cycles, active cycles, and statistical errors are 300,000, 50, 300, and 0.00007, respectively. The calculated results were normalized to the average power density (6.6 MW/m$^3$) of the fuel assembly. The effects of VF and pitch changes on the fast neutron flux distribution in a single fuel assembly (Figure 2) will be discussed separately below.

### 3.1. Influence of VF on the Fast Neutron Flux Distribution

In MSR core design, VF is typically optimized to enhance fuel utilization [22]. The research goal of this paper is to reduce the average fast neutron flux, local fast neutron flux peak, and fast neutron flux peak factor of the graphite block. The fast neutron flux peak factor is defined as the ratio of the local fast neutron flux peak to the average fast neutron flux of the graphite block, representing the gradient of fast neutron flux distribution in the block. A larger gradient in fast neutron flux distribution increases the likelihood of graphite block failure [25], so the factor should be minimized in the MSR core design.

In the fuel assembly, the diameter of the fuel salt channel is varied, whereas the pitch remains constant at 18 cm. The diameter of the fuel salt channel ranges from 2 cm to 15 cm, resulting in corresponding changes in the VF from about 1% to about 63%. Figure 4 illustrates how the averaged fast neutron flux and fast neutron flux peak factor of the graphite block vary with VF. The fine distributions of fast neutron flux in the fuel assembly along the *X*-axis direction are depicted in Figure 5. For the calculations in Figure 5, the mesh sizes are 0.25 cm, 2 cm, and 320 cm in the x, y, and z directions, respectively (these mesh sizes are also used for subsequent fast neutron flux calculations of local graphite blocks).

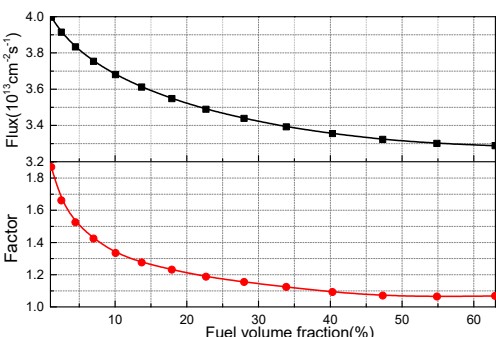

**Figure 4.** Averaged fast neutron flux and fast neutron flux peak factor in the graphite block varying with the VF of the fuel assembly.

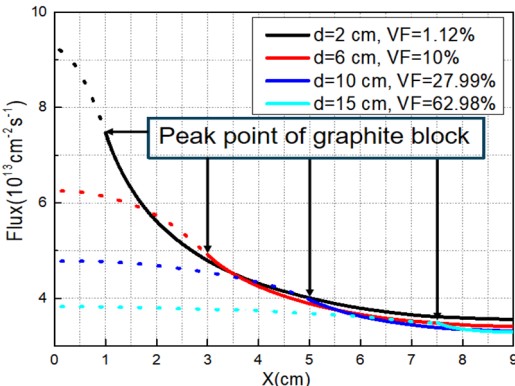

**Figure 5.** Fast neutron flux fine distribution in the fuel assembly (the dashed lines are the fluxes located in fuel salt and the solid lines are the fluxes located in graphite block).

As shown in Figure 4, the averaged fast neutron fluxes and fast neutron flux peak factors of the graphite block decrease with the increase in VF. In Figure 5, the local fast neutron flux of graphite blocks decreases as X increases, which is evidently related to the distance of the fuel salt channel. The local fast neutron flux peaks are close to the fuel salt channel, and these peaks also decrease with the increase in VF.

This is because fission neutrons are produced from the fuel, and the larger volume of the fuel salt has a more significant moderating effect on the fast neutrons, causing the average fast neutron flux of the graphite block to decrease as the volume of the fuel salt increases. Additionally, with the increase in VF, the fission power density decreases (Figure 6), and the neutron flux level in the fuel salt is significantly reduced. Consequently, the local fast neutron flux peak of the graphite block also decreases with the increase in VF. Moreover, the fast neutron flux distribution in the fuel salts becomes flatter with the increase in VF, resulting in a decrease in the fast neutron flux peak factor of the graphite block.

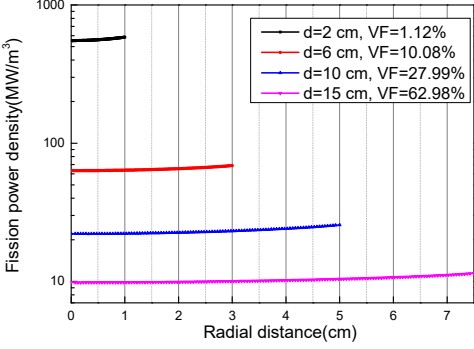

**Figure 6.** Fission power density distribution of the fuel assembly.

### 3.2. Influence of Pitch on the Fast Neutron Flux Distribution

Section 3.1 demonstrates that the VF has a significant influence on the fast neutron flux distribution. However, the actual core design also needs to take into consideration the impact of the assembly size under the same VF. Therefore, it is necessary to study the influence of pitch on fast neutron flux distribution.

The pitch of the fuel assembly (Figure 2) is changed while keeping the VF at 10.08%. Figure 7 displays the averaged fast neutron flux and fast neutron flux peak factor of the graphite block as they vary with the pitch. The fine distributions of fast neutron flux for the fuel assembly in the *X*-axis direction are depicted in Figure 8.

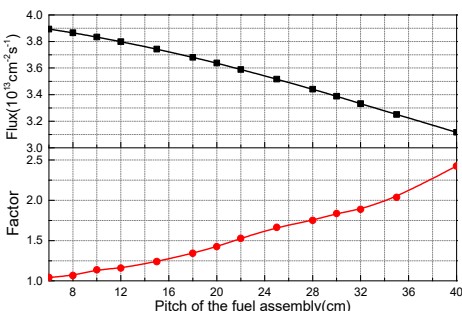

**Figure 7.** Averaged fast neutron flux and fast neutron flux peak factor of the graphite block varies with the pitch of the fuel assembly.

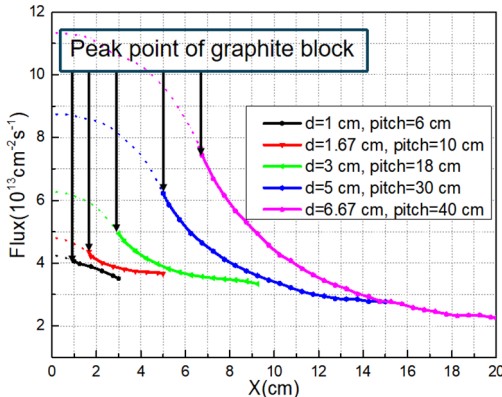

**Figure 8.** Fast neutron flux fine distribution of the fuel assembly (the dashed lines are the fluxes located in fuel salt and the solid lines are the fluxes located in graphite block).

In the figures, the average fast neutron flux of the graphite block decreases as the pitch increases in a regular hexagon fuel assembly. However, the local fast neutron flux peak and fast neutron flux peak factor of the graphite block increase with the pitch.

As the pitch increases, fast neutrons produced in the fuel salt are slowed down more easily by the fuel salt, thereby reducing the probability of fast neutrons leaking into the graphite. Consequently, the average fast neutron flux of the graphite block decreases as the pitch increases. Additionally, the fission power density peak increases with the pitch (Figure 9), and the peak is located close to the graphite block, resulting in an increase in the local fast neutron flux peak of the graphite block. Moreover, the fast neutron flux distribution within the assembly becomes steeper with the pitch increases (Figure 8), causing the fast neutron flux peak factor of the graphite block to increase with the pitch.

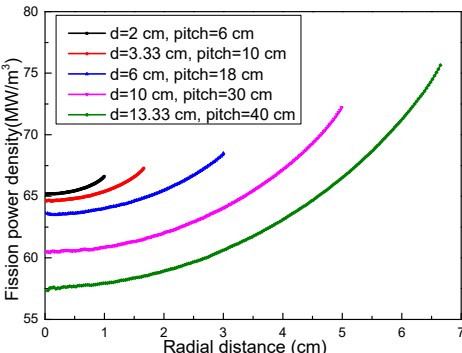

**Figure 9.** Distribution of fission power density in the fuel assembly for pitches of 6 cm, 10 cm, 18 cm, 30 cm, and 40 cm.

From the above calculation and analysis of a regular hexagon fuel assembly, we obtain the following conclusions:

(1) Increasing VF is an effective method to flatten the fast neutron flux distribution of the graphite block in a single fuel assembly. However, for a core, when varying VF, the following also need to be taken into consideration: the impact of fuel utilization, temperature reactivity, and core temperature distribution. A reasonable range of VF needs to be selected through comprehensive evaluation.

(2) Adjusting the pitch of a single fuel assembly, whether increasing or decreasing it, is not an effective choice for flattening the fast neutron flux distribution of its graphite block. The size of the graphite block is generally determined based on mechanical analysis. According to MSBR research [8], the stress of internal irradiation deformation is much lower than the allowable stress when the pitch of the graphite assembly is less than 10 cm. Of course, when varying pitch, the following also need to be taken into consideration: the impact of fuel utilization, temperature reactivity, and heat transfer in the core.

(3) The fission power density distribution has a significant influence on the fast neutron flux distribution. Therefore, flattening the fast neutron flux distribution can be achieved by reducing the fission power density distribution in the inner region and extending the graphite lifespan of the core.

## 4. Calculation Results of Core Optimization

### 4.1. Influence of Core Zoning on the Fast Neutron Flux Distribution

According to the above calculation and analysis of the fuel assembly, changing the VF of the fuel assemblies is an effective way to vary the fast neutron flux distribution. At the core level, the averaged fast neutron flux distribution of graphite blocks in the core mainly depends on the fission power distribution. There are two ways to reduce the fission power distribution of the inner core region. The first is to reduce the fuel salt amount, and the second is to harden the energy spectrum of the inner core region. Taking the configuration VI cores as an example, the neutron energy spectra of the inner core regions change with VF as shown in Figure 10.

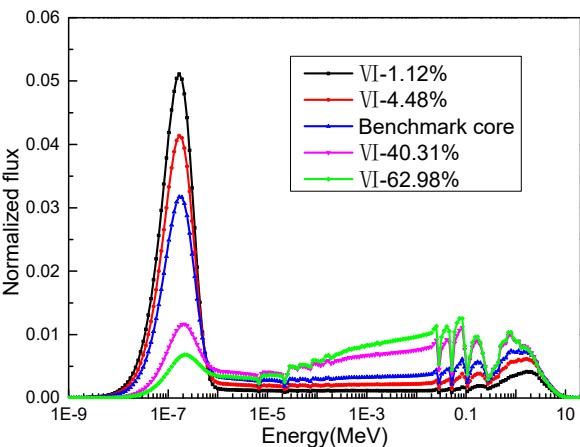

**Figure 10.** Neutron energy spectrum of the inner core regions.

In Figure 10, each energy spectrum plot was created using 200 data points. It is evident from the figure that the neutron spectra vary significantly with the VF. Compared with the benchmark core, the energy spectra of the fuel channels in the VI-40.31% and VI-62.98% inner core regions are hardened, resulting in a decrease in the fission cross-section. We define the mean microscopic fission cross-section of the inner region as $\overline{\sigma}$ as follows:

$$\overline{\sigma} = \int_{\Delta E} \phi(E)_N \sigma(E)_{235} d(E), \tag{1}$$

where $\phi(E)_N$ is the neutron energy spectrum of the inner core region (Figure 10) and $\sigma(E)_{235}$ represents the fission cross-section of $^{235}$U when the neutron incident energy is $E$. The calculated fission amount per unit time in the inner region is shown in Table 3.

**Table 3.** Calculated fission amount per unit time in the inner region.

| VF | VI-1.12% | VI-4.48% | VI-10.08% | VI-40.31% | VI-62.98% |
|---|---|---|---|---|---|
| The mean microscopic fission cross section of the inner region $\overline{\sigma}$ (Barn) | 200.7 | 163.7 | 129.3 | 56.9 | 37.9 |
| Amount of $^{235}$U in the inner region (mol) | 24.3 | 64.8 | 121.5 | 485.9 | 759.3 |
| Fission rate in the inner region (s$^{-1}$) | $1.32 \times 10^{18}$ | $3.21 \times 10^{18}$ | $3.84 \times 10^{18}$ | $3.21 \times 10^{18}$ | $2.10 \times 10^{18}$ |

As seen from the table, the benchmark core (VF = 10.08%) has the largest amount of fission per unit time in the inner region. Decreasing VF (mainly through reduced fuel loading) or increasing VF (mainly through reduced fission cross-section) can reduce this amount. Therefore, decreasing or increasing the VF of the inner core region could flatten the averaged fast neutron flux distribution of the graphite blocks. In the sm-TMSR inner core region, the pitch of the fuel assemblies remains fixed, whereas the diameter of the fuel salt channels is changed. The diameters of the fuel channels considered in the calculation range from 2 to 15 cm (corresponding to VFs of about 1% to 63%).

The variation of VF in the inner region will cause the reactor to deviate from the critical state, which is inconsistent with the actual operation state. If the effective multiplication factor k-eff of the core is less than 1, the molar ratio of the fuel heavy metal (HM) is adjusted to keep the core in a critical state. The moderating ability of HM is relatively weak; although it may affect thermal neutron absorption, it does not influence the distribution of fast neutron flux. Therefore, in the calculation, if the k-eff of the core is less than 1, the HM is adjusted to maintain k-eff at approximately greater than 1. The averaged fast neutron flux distributions of the graphite blocks in the six configuration cores are investigated, as shown in Figure 11.

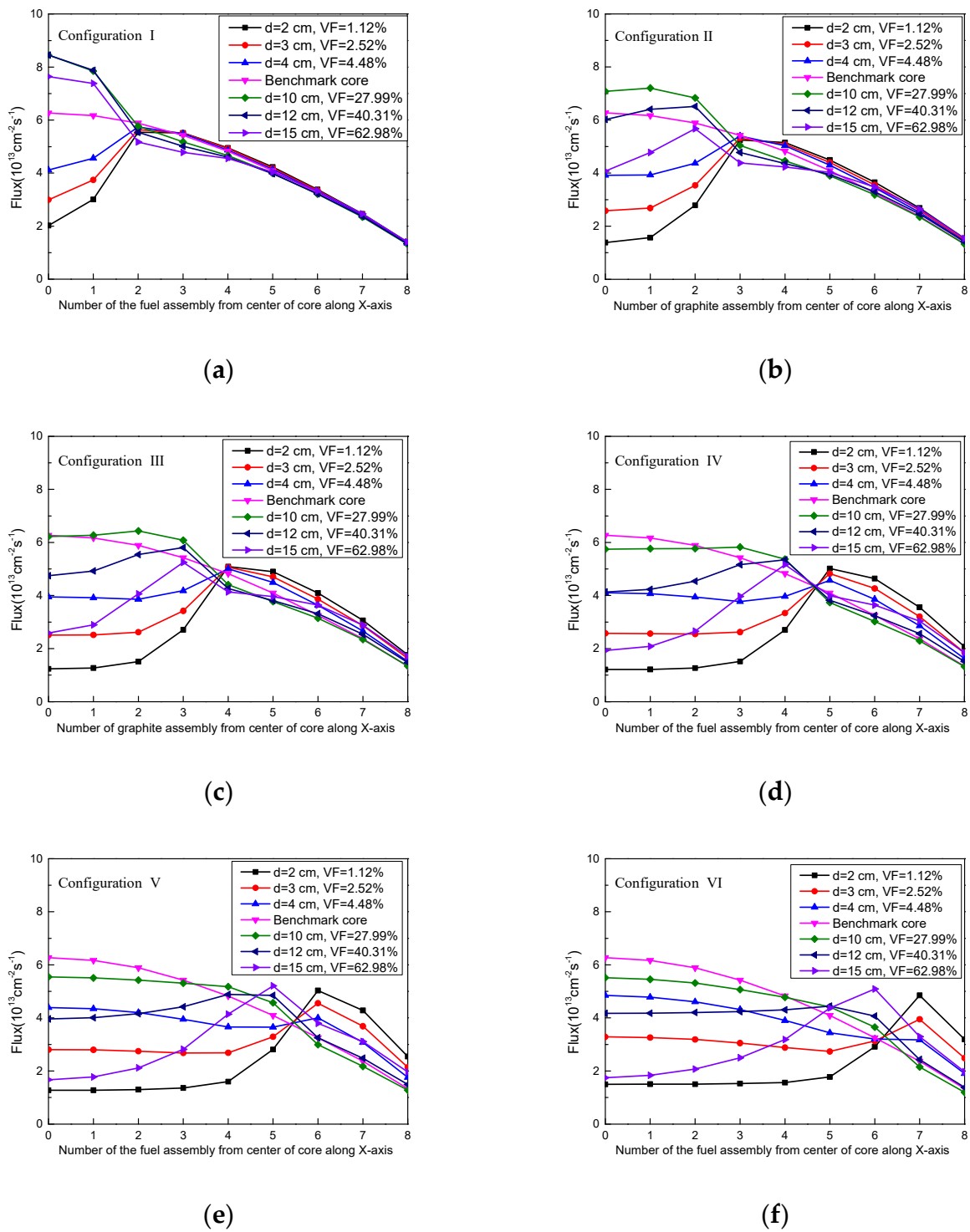

**Figure 11.** Averaged fast neutron flux distributions of the graphite blocks in the *X*-axis fuel assemblies of the sm-TMSR cores. (**a**–**f**) represent the calculation results of the configuration I, II, III, IV, V, and VI cores.

As observed in Figure 11, the averaged fast neutron flux distributions of the graphite blocks notably change with variations in VF across all six configuration cores. In the benchmark core (d = 6 cm, VF = 10.08%), the maximum averaged fast neutron flux of the graphite blocks is located at the center of the core. In configurations I, II, and III, the variation of VF has a greater influence on the fast neutron flux curve in the inner region but less influence on the outer region. Decreasing the VF of the inner core regions can reduce

the maximum averaged fast neutron flux of the graphite blocks, but the flattening effect is not very significant, thus requiring further expansion of the inner region range.

In the configuration IV, V, and VI cores, decreasing or increasing the VF of the inner core regions can also reduce the maximum averaged fast neutron flux of the graphite blocks. Furthermore, it is noted that the cores of the following exhibit a better flattening of the fast neutron flux distribution: IV-4.48%, V-4.48%, VI-2.52%, and VI-40.31% (the roman numeral representing the serial number of configurations as shown in Figure 3 and the % representing the VF of the inner core region). Compared with the benchmark core, the maximum averaged fast neutron flux of the *X*-axis graphite blocks in the , V-4.48%, VI-2.52%, and VI-40.31% cores decrease by approximately 27%, 30%, 47%, and 29%, respectively.

The averaged fast neutron flux of the graphite blocks in the fuel assemblies of the sm-TMSR core are mainly determined by the fission power distributions surrounding them. Taking the configuration IV cores as an example, the averaged fast neutron flux distributions of the graphite blocks and the fission power distributions of corresponding fuel assemblies are shown in Figure 12.

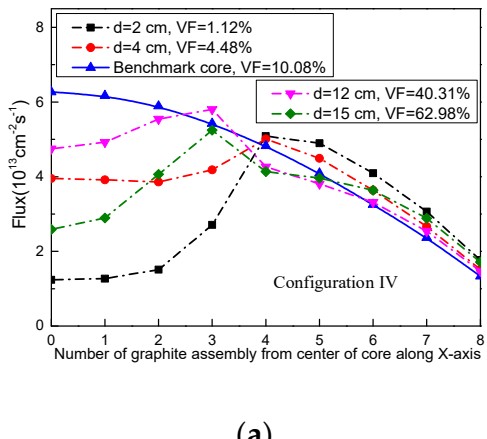 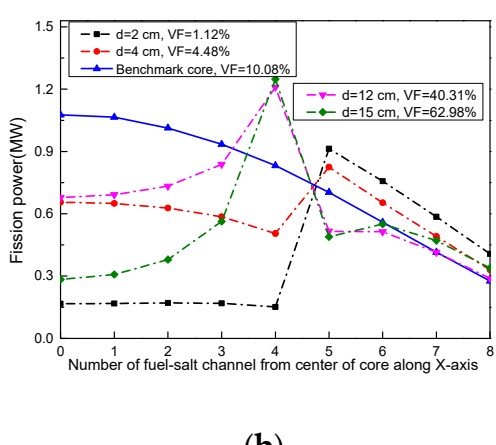

**(a)** **(b)**

**Figure 12.** (**a**) Averaged fast neutron flux distributions of the graphite blocks and (**b**) fission power distributions of corresponding fuel assemblies in the IV-1.12%, IV-4.48%, benchmark, IV-40.31%, and IV-62.98% cores.

As shown in Figure 12, the maximum averaged fast neutron fluxes of the graphite blocks in the IV-1.12%, IV-4.48%, benchmark, IV-40.31%, and IV-62.98% cores are located on the fuel assemblies marked as 5, 5, 0, 4, and 4, respectively, where the fission power peaks of the fuel salt channels are also located. Additionally, the averaged fast neutron flux distributions of the graphite blocks are nearly consistent with the fission power distributions of the corresponding fuel assemblies.

In the benchmark, IV-4.48%, V-4.48%, VI-2.52%, and VI-40.31% cores, we are more interested in the fast neutron flux fine distributions of the *X*-axis fuel assemblies and fast neutron flux peak factors of each graphite block, as shown in Figures 13 and 14, respectively.

As depicted in Figure 13, the local fast neutron flux peaks of the graphite blocks in the benchmark, IV-4.48%, V-4.48%, VI-2.52%, and VI-40.31% cores are situated on the fuel assemblies marked as 0, 5, 0, 7, and 6, respectively. In comparison with the benchmark core, the local fast neutron flux peaks of the graphite blocks in the IV-4.48%, V-4.48%, VI-2.52%, and VI-40.31% cores decrease by approximately 25%, 20%, 32%, and 42%, respectively.

In Figure 14, it is evident that the fast neutron flux peak factors are solely associated with the VF in the center of the core, and the factor diminishes with the increase in VF. Compared with the center of the benchmark core, the fast neutron flux peak factors in the center of the IV-4.48%, V-4.48%, and VI-2.52% cores increase by about 13%, 13%, and 24%, respectively, but decrease by 20% in the center of the VI-40.31% core. Consequently,

the VI-40.31% core is the better choice from the perspective of flattening the fast neutron flux distribution.

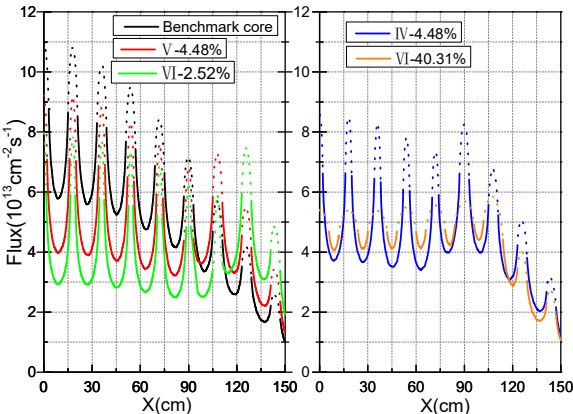

**Figure 13.** Fast neutron flux fine distributions of the *X*-axis fuel assemblies in the benchmark, IV-4.48%, V-4.48%, VI-2.52%, and VI-40.31% cores (the dashed lines are the fluxes located in fuel salt and the solid lines are the fluxes located in graphite block).

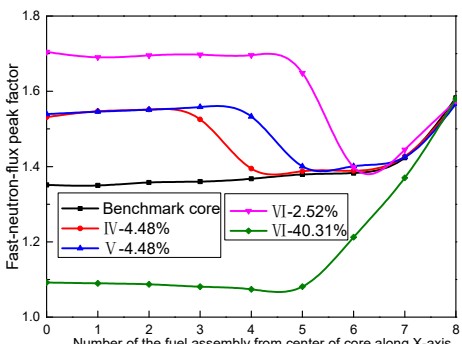

**Figure 14.** Fast neutron flux peak factors of each graphite block in the *X*-axis fuel assemblies of the benchmark, IV-4.48%, V-4.48%, VI-2.52%, and VI-40.31% cores.

### 4.2. Influence of Core Zoning on the Burnup, Initial k-eff, and Temperature Distribution

The neutron economy is an essential aspect to consider in core design. The VF variation of the inner core region directly impacts the initial loading of fuel salt and the neutron energy spectrum, and significantly affects neutron economy, including burnup depth and initial k-eff. Figure 15 displays the burnup depth changing with time in the benchmark, IV-4.48%, V-4.48%, VI-2.52%, and VI-40.31% cores.

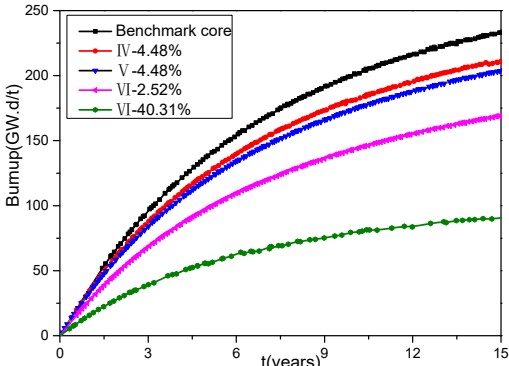

**Figure 15.** Burnup depth changing with time in the benchmark, IV-4.48%, V-4.48%, VI-2.52%, and VI-40.31% cores.

As it can be seen in the figure above, flattening the fast neutron flux distribution is detrimental to the burnup depth of the core. Previous research [12,22] indicates that when VF is approximately 10%, the burnup depth reaches its maximum. In simpler terms, the initial k-eff and burnup depth are at their highest at the transition point between under-moderation and over-moderation. A decrease in the VF of the inner region leads to an increase in both neutron leakage and graphite absorption. Conversely, an increase in the VF results in a hardened neutron energy spectrum, which is also unfavorable for burnup. Table 4 shows the neutron economy of the benchmark, IV-4.48%, V-4.48%, VI-2.52%, and VI-40.31% cores. In Table 4, the initial k-eff and uranium loading are determined using the same fuel composition. For the burnup calculation, since the initial k-eff of the VI-2.52% and VI-40.31% cores are less than 1, the HM is adjusted to maintain criticality, after which the burnup calculation is performed.

**Table 4.** Comparison of the 10(EFPY) burnup depth, initial k-eff, initial uranium loading, and average VF in the active core between the benchmark core, IV-4.48%, V-4.48%, VI-2.52%, and VI-40.31% cores.

| Configuration | 10(EFPY) Burnup Depth (GW.d/t) | Initial k-eff | Initial Uranium Loading in the Active Core (kg) | Average VF in the Active Core ($cm^{-2}s^{-1}$) |
|---|---|---|---|---|
| Benchmark core | 200.1 | 1.04947 | 273.7 | 10.8% |
| IV-4.48% core | 182.1 (−9%) | 1.02109 | 235.2 (−14.1%) | 8.66% |
| V-4.48% core | 174.1 (−13%) | 1.00799 | 216.2 (−21.0%) | 7.97% |
| VI-2.52% core | 143.8 (−28%) | 0.92519 | 165.5 (−39.5%) | 6.10% |
| VI-40.31% core | 80.2 (−60%) | 0.86124 | 706.3 (158.1%) | 26.02% |

From Table 4, it can be observed that, compared to the benchmark core, the 10 (EFPY) burnup of the IV-4.48%, V-4.48%, VI-2.52%, and VII-40.31% cores decreases by approximately 9%, 13%, 28%, and 60%, respectively. Compared to the benchmark core, the VI-2.52% and VII-40.31% cores show very poor neutron economy due to the significant decrease in burnup depth and initial k-eff; as a result, they must be excluded from the optimized core choice.

Considering both the fast neutron flux distribution and neutron economy, the IV-4.48% core is a better choice and is considered an optimized core. The burnup depth, initial k-eff, initial uranium loading, and average VF of the IV-4.48% active core showed a slight decrease compared to the benchmark core. Therefore, the optimized core can be seen as an optimal compromise between neutron economy and the flattening of the fast neutron flux distribution.

The change in core structure not only affects neutron physics but also has a significant influence on core temperature distribution. The outlet temperature distribution, calculated using the single-channel model, is shown in Figure 16 for both the optimized (IV-4.48%) and benchmark cores.

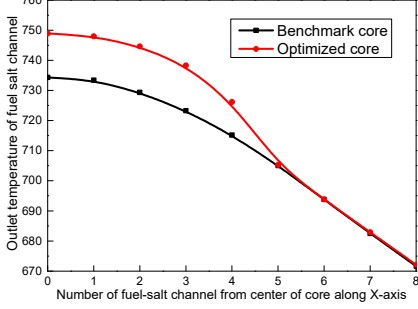

**Figure 16.** Outlet temperature distribution of the fuel channel along the *X*-axis in the optimized and benchmark cores.

In the calculation, the fission power distribution is shown in Figure 12b. It is assumed that the inlet temperature of the core is 650 °C, the average outlet temperature is 700 °C,

and the inlet flow velocity remains constant in each channel within the core. The inlet flow velocities in the benchmark and optimized cores are calculated to be 0.70 m/s and 0.81 m/s, respectively. The mass flow rate per fuel salt channel in the benchmark core is 5.35 kg/s, whereas in the optimized core, it is 2.78 kg/s in the inner region and 6.25 kg/s in the outer region.

Figure 16 illustrates that the temperature distribution in the inner region of the optimized core is slightly higher than that in the benchmark core, attributed to the reduced mass flow rate in the inner region of the optimized core. However, core zoning does not lead to the deterioration of core heat transfer efficiency. By considering heat transfer between the fuel assemblies and the actual mass flow distribution (with a higher inlet flow velocity at the core center), the optimized core outlet temperature distribution is better than the distribution shown in the figure.

### 4.3. Influence of Control Rod Channels Arrangement on the Fast Neutron Flux Distribution

The control rod assembly is an essential cell of the reactor core, and its presence will affect the power distribution, subsequently influencing the fast neutron flux distribution within the core. This may result in local power or fast neutron flux distribution distortion, significantly affecting the lifespan of graphite. Therefore, it is crucial to arrange the control rod assemblies in a reasonable and reliable manner within the core. The structure of the sm-TMSR control rod assembly is shown in Figure 17. Even when the control rod is not inserted into the core, the Hastelloy annular tubes of the control channel exhibit strong neutron absorption. The fast neutron flux distribution can be further flattened by rationally arranging the control rod assemblies appropriately. The impact of the arrangement of control rod assemblies on the fast neutron flux distribution is investigated in the IV-4.48% core.

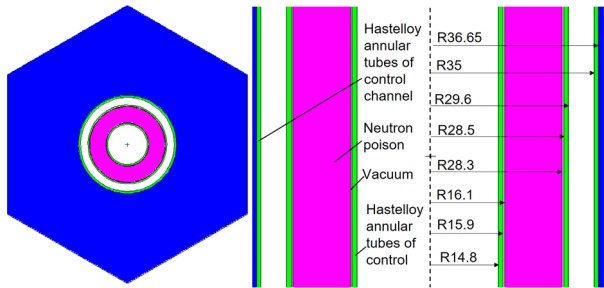

**Figure 17.** Control rod assembly and structure of control rod.

Since the loss of reactivity due to burnup can be compensated by online refueling during the operation of the sm-TMSR, only shutdown and regulating rods are considered in the core. The requirements for the value of control rods [10] are as follows: (1) the total design worth of regulating rods is required to be equal to or slightly greater than about 3000 pcm at the beginning of life (BOL) and about 2500 pcm at 10 (EFPY), respectively; (2) the "one stuck rod" criterion should be satisfied by shutdown rods, and their total design worth is required to be more than about 2000 pcm at BOL. The control rod assemblies are placed at the fifth circle to reduce the maximum averaged fast neutron flux and local fast neutron flux peak of the graphite blocks in the IV-4.48% core. Additionally, the power distribution of the fifth circle is also the largest, and the arrangement of the control rod assemblies in this location is conducive to maximizing the value of the control rods.

As shown in Figure 18, three regulating rods and three shutdown rods are arranged symmetrically at the fifth circle. The calculation results indicate that the total value of the regulating rods is about 4500 pcm, and the value of two shutdown rods is about 2600 pcm at BOL, fully meeting the core reactivity requirements. Figure 19 shows the fast neutron flux distribution in the IV4 core with (the control rod is not inserted into the core) or without control rod assemblies. Figure 20 illustrates the averaged fast neutron flux distributions of the *X*-axis graphite blocks in the IV-4.48% core without and with control rod assemblies.

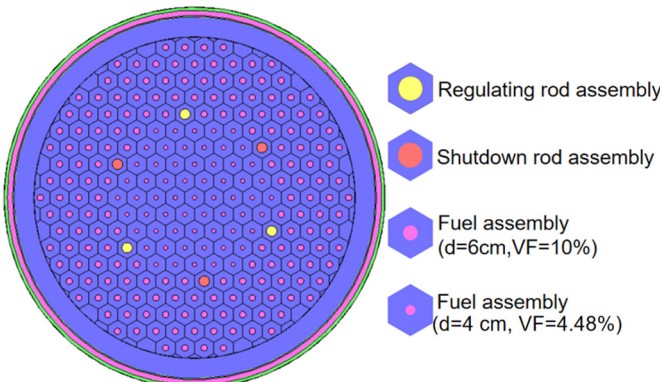

**Figure 18.** Schematic diagram of regulating rods and shutdown rods arrangement in the IV-4.48% core.

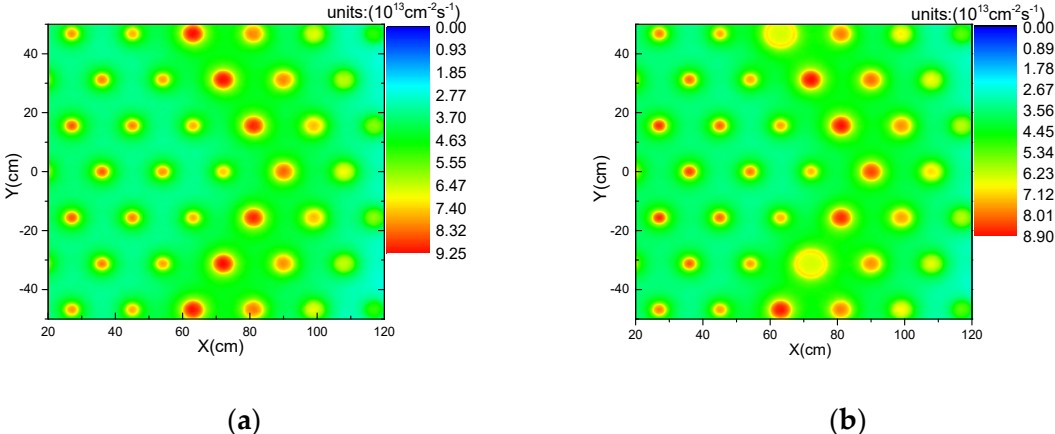

(**a**)                                                      (**b**)

**Figure 19.** Fast neutron flux distributions in IV-4.48% core without (**a**) and with (**b**) control rod assemblies.

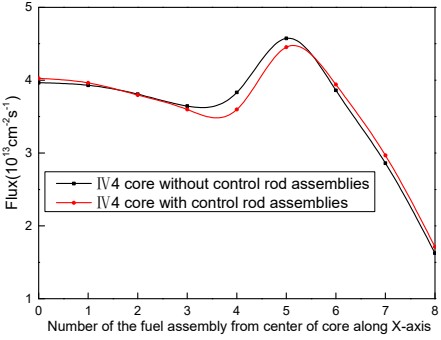

**Figure 20.** Averaged fast neutron flux distribution of the graphite blocks in the IV-4.48% core without and with control rod assemblies.

As seen in the figures, the control rod assemblies influence the fast neutron flux distribution around them. The local fast neutron flux peak of the core is reduced by about 4%, indicating that the fast neutron flux distribution of the core can be further flattened. Both the maximum averaged fast neutron flux and local fast neutron flux peak of the graphite blocks can be reduced by about 3% when the control rod assemblies are symmetrically arranged at the fifth circle of the IV-4.48% core.

### 4.4. The Final Optimized Core

From the perspective of fast neutron flux distribution, burnup, initial k-eff, and reactivity control, the final optimized core was obtained (Figure 18). The inner region of the

optimized core comprises circles 0, 1, 2, and 3, and its VF is equal to 4.48%. Summarized in Table 5, the main parameters of the optimized core are compared with those of the benchmark core.

**Table 5.** The parameters of the benchmark core and the final optimized core.

| Parameters | Benchmark Core | Optimized Core | Percentage Difference |
|---|---|---|---|
| VF of the core | 10.08% | Inner region: 4.48% Outer region: 10.08% | – |
| Maximum averaged fast neutron flux of graphite blocks in the core ($cm^{-2}s^{-1}$) | $6.27 \times 10^{13}$ | $4.45 \times 10^{13}$ | −29.0% |
| Local fast neutron flux peak of graphite blocks in the core ($cm^{-2}s^{-1}$) | $8.47 \times 10^{13}$ | $6.17 \times 10^{13}$ | −27.2% |
| Fast neutron flux peak factors of graphite block in the center of the core | 1.35 | 1.53 | 13.3% |
| Number of core replacements during the lifetime of the reactor | 6 | 4 | −33.3% |
| 10(EFPY) burnup depth of the core (GW·d/t) | 200.1 | 166.0 | −17.0% |
| Fuel salt temperature coefficient at BOL(pcm/K) | −4.0 | −4.3 | 7.5% |
| Graphite salt temperature coefficient at BOL(pcm/K) | −3.2 | −4.8 | 37.5% |
| Total temperature reactivity coefficient at BOL (pcm/K) | −6.8 | −8.6 | 26.5% |
| Fuel salt temperature coefficient at 10(EFPY) (pcm/K) | −0.4 | −0.5 | 25.0% |
| Graphite salt temperature coefficient at 10(EFPY) (pcm/K) | −1.2 | −2.2 | 83.3% |
| Total temperature reactivity coefficient at 10(EFPY) (pcm/K) | −1.4 | −2.5 | 78.6% |

In Table 5, compared with the benchmark core, although the fast neutron flux peak factors of graphite block in the center of the core increased by 13.3% and the 10(EFPY) burnup depth of the core with control rod assemblies was reduced by 17.0%, the maximum averaged fast neutron flux and local fast neutron flux peak of the graphite blocks in the core decreased significantly. Based on a permissible fast neutron exposure value ($3 \times 10^{22}$ n/cm$^2$) [16] for the MSR graphite block divided by the local fast neutron flux peak of the graphite blocks, the optimized core life is about 15 years. The optimized core results in two fewer graphite replacements over the life of the reactor. Moreover, the optimized core ensures a larger negative temperature reactivity coefficient. Reasons for the decrease in burnup can be attributed to the reduction of the fuel salt volume in the inner region and the arrangement of control rod assemblies with alloy sleeves in the optimized core.

## 5. Conclusions

In this study, we address the flattening of the fast neutron flux distribution to prolong the graphite irradiation lifespan in the commercial sm-TMSR core. Several valuable conclusions are obtained and summarized as follows:

(1) At the fuel assembly level, changing the VF of the fuel assembly is more effective in flattening the fast neutron flux distribution than varying the pitch.

(2) At the core level, the effect of zoning the core into two regions on the fast neutron flux distribution flattening was studied. The research revealed that appropriately reducing or increasing the VF of the inner core region can effectively flatten the fast neutron flux distribution, with VF being increased proving more effective than VF being reduced.

(3) However, core zoning would clearly impact neutron economy. Considering the fast neutron flux distribution and burnup, an optimized core selects a configuration that appropriately reduces the VF of the inner region. Based on this core, the fast neutron flux distribution is further optimized by arranging the control rod assemblies reasonably, resulting in an additional reduction of approximately 3% in the fast neutron flux.

(4)     Compared with the benchmark core, the final optimized core effectively reduces both the maximum averaged fast neutron flux and the local fast neutron flux peak of the graphite blocks. Over the life of the reactor, the number of optimized core replacements has decreased by two, while maintaining a larger negative temperature reactivity coefficient, a relatively minor reduction in burnup, and a fully acceptable core temperature distribution.

**Author Contributions:** Conceptualization, G.Z.; methodology, X.K.; software, G.Z.; validation, G.Z., Y.L. and J.W.; formal analysis, G.Z. and X.K.; investigation, X.K.; resources, R.Y. and Y.Z.; data curation, X.K.; writing—original draft preparation, X.K.; writing—review and editing, X.K., G.Z., J.W., R.Y., Y.Z. and Y.L.; visualization, X.K., J.W. and Y.L; supervision, R.Y. and Y.Z.; project administration, R.Y. and Y.Z.; funding acquisition, Y.Z. All authors have read and agreed to the published version of the manuscript.

**Funding:** This work was funded by [Guifeng Zhu] grant number [No. 12005290, No. 2020261, JCYJ-SHFY-2021-003] and [Jianhui Wu] grant number [No. 11905285, No. 20ZR1468700].

**Data Availability Statement:** The original contributions presented in the study are included in the article, further inquiries can be directed to the corresponding author.

**Acknowledgments:** We, the authors, would also like to extend our gratitude to Jinhong Zhang for providing fruitful comments and discussions in the preliminary thermal analysis.

**Conflicts of Interest:** The authors declare no conflicts of interest.

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
