# Peer review of "Core Optimization for Extending the Graphite Irradiation Lifespan in a Small Modular Thorium-Based Molten Salt Reactor"

_jne, doi:10.3390/jne5020012_

Round 1

Reviewer 1 Report

Comments and Suggestions for Authors

This is an interesting study with a certain level of application potentials for thorium based MSR design. Some minor revisions are suggested as follows, to give more clarification and contribute some improvements to the current manuscript:

 (1). Title: the word ‘thorium’ is worthy of mentioning in the title, and conclusions or indications could be quite different if the MSR is based on uranium fuels.

(2). In neutronics analysis, the fuel volume fraction (VF) and fuel temperature coefficient (FTC) is normally optimized in the sense of effective multiplication factor (k-eff) as these factors would greatly affect the k-eff performance, and we are certainly desired to achieve the best k-eff with least the fuel inventory. In this regard, it is a big concern to me if all the results or design outcome presented in this paper affect the k-eff values as no intention to present these results at all because the whole point of this study is to flatten the fast-neutron-flux among the sm-TMSR core. This is understandable as this is main study goal of this work, however, clarifying the effect on k-eff is still very important, because if the best design that achieves the best fast-flux distribution but meanwhile have an acceptable k-eff performance, it is surely needed to be abandoned in my opinion.

(3). Check though and use the consistent term ‘sm-TMSR’ all the times, as somewhere I see ‘smTMSR’, or vice versa.

(4). Page 1, line 42, it appears the cutoff energy 0.05MeV is defined for fast neutron flux, is this value you use to tally the fast flux in all your Monte Carlo calculations? If so, this need to be clarified during the calculation, as 0.05 MeV is not a typical cutoff energy we define fast energy group.

(5). Page 2, line 63-79, this paragraph summaries a variety of MSR designs with different VF and fuel cycle considerations, would it be more indicative if producing a table for this summary, or you may add more columns in Table 1 for these information, as they are basically discussing the same types of reactors.

(6). Page 3, Section 2, I suggest to start with the introduction of  assembly model, and then the core model, as your calculation procedure and result presented in Section 3 and 4 are in this order.

(7). Page 3, Figure 1, larger fonts for the labels shown in the figure are desired: the current ones shown in the figure are nearly invisible.

(8). Page 3, Table 2, no split into pages for tables and figure (+figure captions)

(9). Page 3, Table 2, some important information are missing for the core design parameters, including # of assemblies used, thickness of reflector, thickness of vessel, and thickness of downcomer. Anyhow, if you expect others to be able to reproduce your results, these are needed parameters.

(10).        Page 3, Table 2, the active core size is noted in the units of mm, some all other length dimensions used in the text are in cm or m. You should consider all consistent units all through the manuscript.

(11).        Page 4, Table 3, what’s the difference of ‘design of the reactor’ and ‘design of the core’? To remove ambiguity of these terms, they should be explained in relevant context.

(12).        Page 4, Figure 2, does the design really have a assembly as shown in Figure 2? Or it is really an imaginary one just for easy computation simulation purpose. I would be surprised if the core is actually constructed by forming many assemblies as such to leave possible gap space between assemblies.

(13).        Page 4, Section 2.2, it is confusing to call it with a subject as ‘method of calculation’ as there is really no method involved, what I see here is just a bunches of different two-region design ideas.

(14).        Page 4, line 138, you should probably explain a bit on ‘specular reflection’ as it may not be easy to follow for people not familiar with this definition.

(15).        Page 4-5, Figure 3, what is main point to end up with the assembly level calculations with 6 different models? What is the logic sense behind here? In my view, it is fairly enough to have 3 of them as the whole point is to see which trend would be favored to flatten the fast neutron flux, correct?

(16).        Page 5-16, Section 3 and Section 4. I am all good with these studies and results presented, however, I would like to repeat again as an reiteration of my point 2, the k-eff results should be examined for all these case studies to justify that these optimization efforts does not hurt neutronics performance as a whole.

(17).        Page 9, Figure 10, how many groups of data you used to generate the energy spectra plot?

(18).        Page 16, Section 5, with the points given in item 2 and 16, I have reserved option to fully trust the results and conclusions made in this section.

Comments on the Quality of English Language

none

Author Response

Dear reviewer,

Thank you for your valuable comments and suggestions on our manuscript. Based on your feedback and requests, we have made modifications to the original manuscript. Additionally, we have addressed your questions point by point. Please see the attachment.

Best regards,

Zhu GuiFeng

April 20, 2024

Reviewer 2 Report

Comments and Suggestions for Authors

A few minor issues throughout:

-Section 2.2, line 135: I assume you're referring to the KENO module in SCALE (i.e., you refer to the Monte Carlo program SCALE, however KENO is the module which performs Monte Carlo transport analysis in SCALE, which is a larger collection of packages).

-Lines 200-202: You state that the fast neutron flux decreases as the volume fraction of fuel salt increases due to the moderation of the fuel salt. This seems unlikely and is contradicted by line 301.

-Lines 236-238; this claim seems to be contradicted by Figure 8.

-Line 251: would be useful to call out Figure 6 here once more for context.

-Line 259: Would be useful call out Figure 7 once more here for context.

-Line 272: You cite Figure 12 before Figures 10 or 11; either move it up in the text (to make this Figure 10) or hold off on discussing it. I'm also not sure I see how Figure 12 relates to the point being made.

-Lines 301-302: I agree with the conclusion here but it contradicts what is stated previously.

Comments on the Quality of English Language

- Throughout the paper, you refer to "burn-up depth" several times but it is unclear to what you are referring. Are you simply referring to the magnitude of the total burnup?

- Table 3: I would condense the descriptions used in the table. (e.g., "fission rate" instead of "amount of fission per unit time")

- Plural form of spectrum is "spectra"

- Line 299-300: "Adjusting the mole ratio..." sentence fragment, consider revising. ("Adjusting the molar ratio..." is necessary? ...)

- Lines 470-471: Suggest, "The optimized core results in 2 fewer graphite replacements over the life of the reactor."

Author Response

Dear reviewer,

Thank you for your valuable comments and suggestions on our manuscript. Based on your feedback and requests, we have made modifications to the original manuscript. Additionally, we have addressed your questions point by point. 

Best regards,

Zhu GuiFeng

April 20, 2024

Round 2

Reviewer 1 Report

Comments and Suggestions for Authors

I don’t feel my first round review comments/questions are all properly addressed. In particular, some comment responses are not well accepted which makes me feel very skeptical to some key results presented in this paper, so I reserve my agreement to publish this paper in the current shape, and claim it should not be published unless serious concerns are clarified.

Major ones:

The only major issue is related to Reply 2, 16, and 18. We all know k-eff is related to burnup, but the k-eff at the initial design won’t be affected by burnup but it will be heavily affected by the geometry and size setting, as well as the fuel to moderator ratio, etc. So if the optimization process targeting best fast-neutron-flux flattening overlooked k-eff by setting different configurations, I believe there is a serious pitfall here, and the best performance core obtained in this paper may not be the best neutronics (neutron economic) core at all. Plus, some responses in this regard are very confusing. In Reply 2, the authors state “In the process of core optimization, if the effective proliferation factor of the core is less than 1, the molar ratio of the fuel heavy metal (HM) is adjusted to maintain the core in a critical state”, what does this mean? So your optimization is actually performed with different amount of fuels? If so what is the point of neutronics optimization. With this said, I had a feeling the authors may neglect the best configuration of k-eff design, but focus on the best flattened fast flux as the main goal is to reduce the irradiation effect on graphite, which of course will end up not the best neutron economic core design. Even so, it is an acceptable design if the sacrifice is not significant, but the current manuscript just completely ignore the k-eff performance, and thus we have no idea how good or bad the finalized ‘optimal core’ in terms of k-eff compared to other abandoned options.

Minor ones:

(1)   Reply 5: did you take actions in the revised paper to address this concern?

(2)   Reply 7: what are the requirements of the journal in this regard? If the review could not view these fonts well, so do other readers. How could you expect the editor to adjust the font size in a figure?

(3)   Reply 12: this is confusing answers. So is the assembly design for computational simulation purpose or for actual engineering purpose? I don’t understand if the space between assemblies will cause some uncertainty (as stated by response), why not get rid of the assembly but use a large chunk of graphite to replace all the assembly design in the core?

(4)   Reply 17: did you add this information into the revised paper? There would be no point to the comment if just answering the question in the author reply letter without making any changes to the manuscript.

Author Response

Dear Reviewer,

Thank you for your valuable comments and suggestions once again on our manuscript. Based on your feedback and requests, we have made modifications to the original manuscript. Additionally, we have addressed your questions point by point. Please see the attachment.

Best regards,

Zhu GuiFeng May 4, 2024

Reviewer 2 Report

Comments and Suggestions for Authors

The authors' revisions are appreciated and serve to clarify the paper.

I have two minor comments:

- Line 210, the reference is missing

- Line 287, I think you mean "effective multiplication factor" (not proliferation factor).

Comments on the Quality of English Language

Some minor grammatical issues but nothing seriously impairing the reading of the paper

Author Response

(The authors gave the same response as above.)

Round 3

Reviewer 1 Report

Comments and Suggestions for Authors

I think all my comments are properly addressed in the R3 version manuscript. I don't have any further suggestions and agree to accept it for publication.